# Impact of Urban Form on $CO_2$ Emissions under Different Socioeconomic Factors: Evidence from 132 Small and Medium-Sized Cities in China

**Ran Guo [1,2], Hong Leng [1,2], Qing Yuan [1,2],* and Shiyi Song [3]**

[1] School of Architecture, Harbin Institute of Technology, Harbin 150006, China; 17b934034@stu.hit.edu.cn (R.G.); hitlaura@hit.edu.cn (H.L.)

[2] Key Laboratory of Cold Region Urban and Rural Human Settlement Environment Science and Technology, Ministry of Industry and Information Technology, Harbin 150006, China

[3] School of Architecture, Xi'an University of Architecture and Technology, Xi'an 710055, China; songshiyi@xauat.edu.cn

* Correspondence: hityuanqing@hit.edu.cn

**Abstract:** The accurate estimation of the impact of urban form on $CO_2$ emissions is essential for the proposal of effective low-carbon spatial planning strategies. However, few studies have focused on the relationship between urban form and $CO_2$ emissions in small and medium-sized cities, and it is especially unclear whether the relationship varies across cities with different socioeconomic characteristics. This study took 132 small and medium-sized cities in the Yangtze River Delta in China to explore how urban form affects $CO_2$ emissions, considering the socioeconomic factors of industrial structure, population density, and economic development level. First, nighttime light data (DMSP-OLS and NPP-VIIRS) and provincial energy data were used to calculate $CO_2$ emissions. Second, four landscape metrics were used to quantify the compactness and complexity of the urban form based on Chinese urban land-use data. Finally, panel data models were established to analyze whether and how different socioeconomic factors impacted the relationship between urban form and $CO_2$ emissions. The results showed that the three socioeconomic factors mentioned above all had obvious influences on the relationship between urban form and per capita $CO_2$ emissions in small and medium-sized cities. The effect of compactness on per-capita $CO_2$ emissions increased with a rise in the proportion of the tertiary industry, population density, and per-capita GDP. However, compactness shows no effects on per-capita $CO_2$ emissions in industrial cities and low-development-level cities. The effect of complexity on per-capita $CO_2$ emissions only increased with the rise in population density. The results may support decision-makers in small and medium-sized cities to propose accurate, comprehensive, and differentiated plans for $CO_2$ emission control and reduction.

**Keywords:** urban form; $CO_2$ emissions; socioeconomic factors; panel data model; small and medium-sized cities





## 1. Introduction

Global warming caused by excessive emissions of greenhouse gasses has led to a series of social and environmental problems that pose a severe threat to humanity. There is unequivocal scientific evidence to conclude that cities contribute 71–76% of global carbon dioxide ($CO_2$) emissions from energy activities, so they are the major focus of $CO_2$ emission mitigation [1,2]. In recent decades, China has achieved remarkable economic growth. However, rapid development has unavoidably led to massive $CO_2$ emissions. China became the country that emits the highest amount of $CO_2$ emissions in the world in 2006 [3]. For this reason, the Chinese government has set a new target for mitigating greenhouse gas emissions by 60–65% by 2030 [4,5] and has pledged to be carbon neutral by 2060 [6]. Based on the current situation, China's emission-reduction task is arduous.

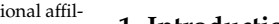

Given the high hazard potential of $CO_2$ emissions, researchers and policymakers have focused on measures to reduce $CO_2$ emissions effectively. Economic restructuring, low-carbon lifestyles, and technological innovations are the conventional methods that have been considered [7,8]. Recently, spatial planning has attracted considerable attention, and proper spatial planning is increasingly used as a measure for addressing climate change and controlling $CO_2$ emissions [9–11]. As a significant aspect of spatial planning, urban form optimization is the main way in which policymakers and planners make interventions. Different urban forms may give rise to diverse social, economic, and ecological consequences [12,13]. Urban form can affect the structure of land use, the construction of infrastructure, urban transportation, urban heat island effects, and carbon sinks, ultimately affecting energy consumption and $CO_2$ emissions in direct and indirect ways [14,15]. At the same time, although urban form has a gradual effect on $CO_2$ emissions, the impact process lasts for a long time since urban form does not usually change rapidly once formed. An undesirable form will have significant socioeconomic and environmental impacts [16]. Previous studies have suggested that under the same conditions, the energy consumption of the ideal urban form is 50–60% less than that of other urban forms [9]. In addition, it has also been found that the reduction of $CO_2$ emissions by conventional methods may not be sufficient to offset the increase in $CO_2$ emissions caused by undesirable urban forms [8]. Thus, a better understanding of the relationship between urban form and $CO_2$ emissions is critically important for ensuring long-term low-carbon development and regional sustainability.

Urban form is the spatial mapping of the interaction between human socioeconomic activities and the natural environment in urban areas [17]. Urban forms can be divided into three categories according to scale: micro, meso, and macro [16,18]. Progress has been made in exploring the relationship between urban form and $CO_2$ emissions at these three scales (hereafter referred to as the UF–$CO_2$ relationship). The micro-scale mainly focuses on the design and structure of buildings and their position with respect to neighborhood buildings and open spaces [19,20]. The meso-scale focuses on the structure and layout of neighborhoods, blocks, lots, open spaces, and streets [20,21]. The macro-scale focuses on the overall structure of the city and development type, including urban complexity, urban compactness, mononuclear patterns, and multiple-nuclei patterns [2]. Given that the scale and spatial structure of small and medium-sized cities is still changing, this study focused on the macro-scale urban form. In addition, according to different research scales, the selection of measurement indicators and data sources on urban forms also varies [22]. As a significant analysis method of conducting landscape ecology, landscape metrics have been widely applied for quantifying macro-scale urban forms [23]. These methods use remotely sensed imagery or land-use data to quantitatively characterize the spatial topology of urban forms by extracting the type, number, distribution, and configuration of the constituent units in the urban landscape [22,24]. Based on the definition of landscape metrics, urban compactness is interpreted as the degree of dispersion or sparseness of the urban land patches. Urban shape complexity is interpreted as the degree of irregularity of urban land patches. A multiple-nuclei pattern describes a city consisting of a group of cores of even distribution of importance [25,26]. If a city has an obvious large urban land patch, which is much bigger than its second-largest patch, it is interpreted as a mononuclear pattern [15,27].

As the world's largest developing and transitioning country, China has been considered an ideal experimental subject for exploring the UF–$CO_2$ relationship, as it is experiencing remarkable expansion in urban scale and energy consumption [28]. Although many studies have been conducted on the macro-scale UF–$CO_2$ relationship, the research conclusions have not been consistent. Taking urban form compactness as an example, it is widely recognized that a compact urban pattern reduces potential transportation requirements and promotes the use of public transport and pedestrian-oriented travel modes; achieves housing energy savings by sharing infrastructure and ensuring efficient use of facilities; and reduces heating-associated energy consumption in buildings by affecting the heat island effect [14,15,29–31]. However, some studies have suggested that compact policies may not

be a sustainable mode for achieving $CO_2$ reductions. Li et al. observed that congestion was positively associated with the degree of compactness and led to an increase in $CO_2$ emissions per kilometer traveled [32]. Ye et al. also found that green infrastructure was usually far away and large in compact cities, resulting in the loss of the opportunity to adjust high ambient temperatures through nearer and more dispersed green infrastructure in densely built-up areas that would save on cooling energy usage [33]. Therefore, based on the conflicting findings, it is reasonable to speculate that these differences in the UF–$CO_2$ relationship might be related to cities with different characteristics (urban scale, climatic conditions, population size, etc.). However, the above studies are mainly case studies in certain cities. It is difficult to draw general conclusions on the UF–$CO_2$ relationship for a particular type of city. Recent studies have begun to systematically consider how the UF–$CO_2$ relationship varies across cities with different characteristics [7,34–36]. Ou et al. sorted 282 prefecture-level cities into five development levels based on a comprehensive system and found that urban forms at different development levels had different impacts on $CO_2$ emissions. Fragmented patterns of urban land can result in higher $CO_2$ emissions for all city tiers. Compact and centralized development does not reduce emissions in tier-one cities [7]. A study of 264 prefecture-level cities by Shi et al. observed that urban form irregularity had a more significant impact on $CO_2$ emissions in cities with an applied low-carbon policy than in regular cities. The $CO_2$ emissions in industrial cities are affected by compactness and complexity, whereas the $CO_2$ emissions of service cities are only affected by complexity [28]. However, such studies have mainly been focused on megacities and large cities in China [37], and few studies have systematically considered the differences in the UF–$CO_2$ relationship in different types of small and medium-sized cities.

According to China's current urban classification standard, cities with a permanent resident population of less than 1 million are considered small and medium-sized cities (here, cities usually refer to counties, which are the third level of administrative division of China). As of 2018, there were 2111 small and medium-sized cities in mainland China [38]. Their population accounts for approximately 75% of China's total population while producing approximately half of China's $CO_2$ emissions [39], with great potential for $CO_2$ emission reduction. However, there are some common problems with reducing $CO_2$ emissions in small and medium-sized cities in China, such as the lack of innovation in the top-level design, the lack of details in $CO_2$ emissions reduction approaches, and the statistical systems of energy consumption have not been established [19,40]. At present, the relative policies are mostly focused on adjusting energy structures, industrial upgrading, and green building demonstration projects. Meanwhile, it is worth noting that China has a vast territory, and there are obvious differences between small and medium-sized cities in terms of their climate background, economic development, population size, industrialization level, and $CO_2$ emission reduction policies [41,42]. Climate conditions have a significant impact on building form and building operation energy consumption [42]. China's building climate zoning includes seven main climate regions (Code for Design of Civil Buildings, GB50352-2005). Small and medium-sized cities are distributed throughout all seven climate regions. In terms of socioeconomic aspects, according to data by Wang et al. in 2016, among small and medium-sized cities, the standard deviations of GDP and population size were 8.64 and 161.17 [43]. Meanwhile, when the industrialization level is measured by the proportion of the added value of the secondary industry to the GDP, the small and medium-sized cities with the highest industrialization level scored 68 times higher than the lowest in 2019 [44]. In recent years, the absolute differences between small and medium-sized cities have been increasing year by year [45]. In 2019, some cities continued to maintain a high economic growth rate of more than 9%, but that of some cities is far lower than the national average of 6.6%, and some cities even have negative growth [38]. Obviously, significant differences in urban socioeconomic characteristics are likely to lead to the UF–$CO_2$ relationship being complex and diverse. It is necessary to explore the impact of urban form on $CO_2$ emissions under different socioeconomic characteristics in small and medium-sized cities to make urban planning decisions in different cities.

The Yangtze River Delta (YRD), as one of the most economically prosperous regions in China, has become a development model for others. Since the reform and opening up in China, the YRD has experienced extremely rapid urbanization [46,47]. Great changes have taken place in terms of urban scale and spatial structure [48], and energy consumption and $CO_2$ emissions intensity have increased significantly [49], which seriously threatens the sustainable development of the region. In addition, most of the study area lies in the hot summer and cold winter climate region, and cities in the same urban agglomeration have similar planning policies and development strategies according to overall regional planning, which makes it an ideal object for our research needs. Thus, in this study, 132 small and medium-sized cities in the YRD were chosen as study subjects to evaluate the impact of urban form on $CO_2$ emissions in small and medium-sized cities under different socioeconomic conditions. Specifically, this study aimed to explore whether and how different socioeconomic factors impact the UF–$CO_2$ relationship in small and medium-sized cities in China, including industrial structure, population density, and economic development level. This study provides a reference for decision-makers and essential guidance for urban planners regarding sustainable urban planning.

## 2. Study Areas

The YRD is located on the eastern coast of China, where an alluvial plain was formed by the flow of the Yangtze River and the Qiantang River into the sea [50]. It consists of one municipality (Shanghai) and three provinces (Jiangsu, Zhejiang, and Anhui) (Figure 1) [51,52]. As mentioned above, cities with a permanent resident population of less than 1 million are considered small and medium-sized in China. However, China's current statistical system is mainly based on administrative divisions as statistical units and collects data on the registered population rather than the permanent resident population. Thus, this study referred to the China Small and Medium-Sized City Development Report (CCDMSC, 2019) to select small and medium-sized cities [53]. According to the classification standard of the report, the YRD includes 167 small and medium-sized cities in total. After removing cities with missing data, 132 cities were chosen as the study objects. Meanwhile, several cities were selected to illustrate the spatial changes in urban land (Figure 2). Data on the cities were collected at 5-year intervals from 2005 to 2020.

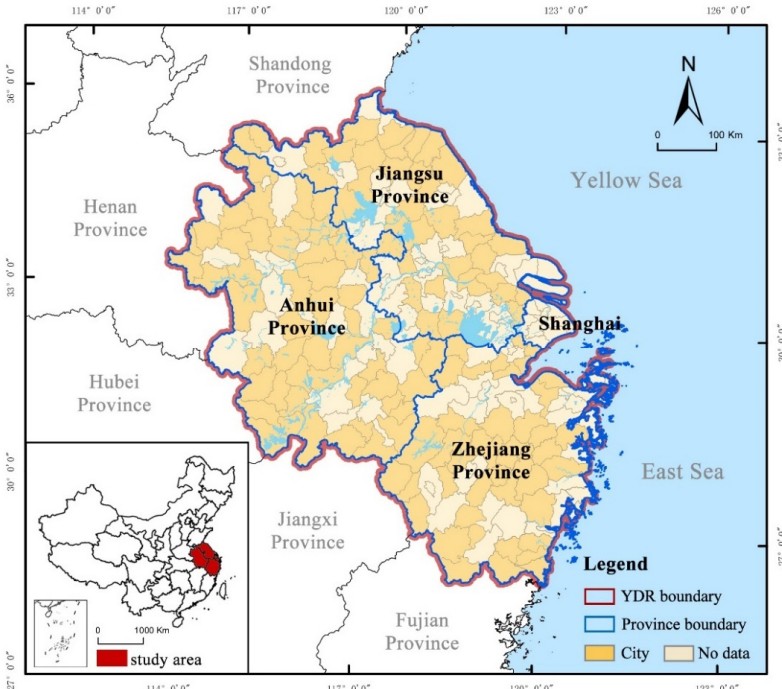

**Figure 1.** The location of the study area in the YRD.

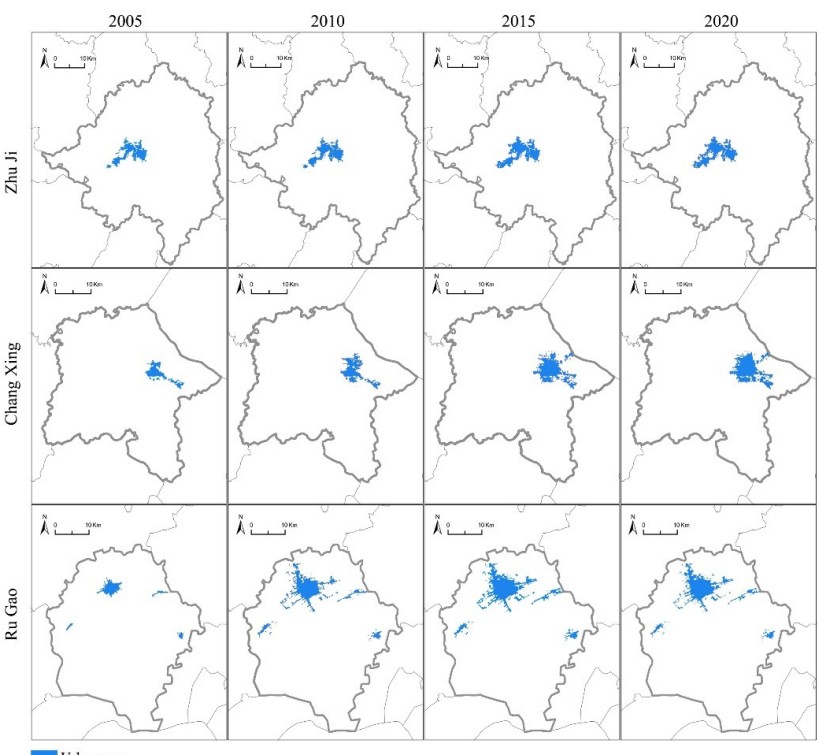

**Figure 2.** The spatial changes of the urban land of typical cities in the YRD from 2005 to 2020.

## 3. Data and Method

### 3.1. Calculating $CO_2$ Emissions

Most of the province-level and city-level $CO_2$ emission data in existing studies were calculated based on the Intergovernmental Panel on Climate Change (IPCC) territorial-based emissions accounting method. Scholars use the energy balance tables of the industrial sectoral energy consumption to calculate the emission inventories of study subjects, which means the total $CO_2$ emissions from one society (within an administrative territory) includes the emissions from different fossil fuels and all sectors. However, as the IPCC method is a bottom-up approach [21], the demand for fossil fuel consumption data is extremely high [22,23]. China's National Bureau of Statistics only discloses the energy consumption data of provinces in its annual China Energy Statistical Yearbook [54], but refined scale energy consumption data at the prefectural level or the small or medium-sized city level are not included. It is difficult to evaluate the $CO_2$ emissions of small and medium-sized cities using the IPCC method [55].

With the development of remote sensing technology, nighttime light data, especially DMSP-OLS and NPP-VIIRS, have been widely and successfully used to estimate $CO_2$ emissions at different city scales since they can directly reflect energy consumption and $CO_2$ emissions associated with social and economic activities [56–58]. However, DMSP-OLS data are available for the period 1992–2013, and NPP-VIIRS data are available from April 2012 onwards, which means no single dataset completely covered the study period. Previous studies have suggested that combining and harmonizing the DMSP-OLS and NPP-VIIRS data to form an extended temporal coverage dataset and using the integrated datasets to estimate the $CO_2$ emissions of small and medium-sized cities were feasible and effective [59–61]. Thus, on the base of previous research, this study combined these two nighttime light datasets to estimate $CO_2$ emissions from 2005 to 2020.

First, the two datasets were corrected following the methods proposed by Liu et al. [62], Shi et al. [63], and Zhu et al. [64] to improve the data accuracy. Second, according to the fitting models of Du et al. [59], this study selected the linear regression model fitting the total digital number values (TDN) of DMSP-OLS and NPP-VIIRS data in 2012 and 2013.

Figure 3 shows that a strong correlation between two datasets and the high coefficient of determination confirms that the model can explain most variations in the regression ($R^2 = 0.949$, $p < 0.01$). Therefore, this regression model was used to correct the NPP-VIIRS data for 2014–2020 and generate the integrated nighttime light dataset from 2005 to 2020. Finally, since the correlation between digital number values and statistical $CO_2$ emissions at the provincial level was consistent with that at the small and medium-sized city level, this study used the linear regression model to quantify the relationships between the digital number values and statistical $CO_2$ emissions at the province level and then estimated the $CO_2$ emissions of the 132 small and medium-sized cities in the YRD from 2005 to 2020. Among them, province-level $CO_2$ emission data were calculated based on the IPCC territorial-based emissions accounting method.

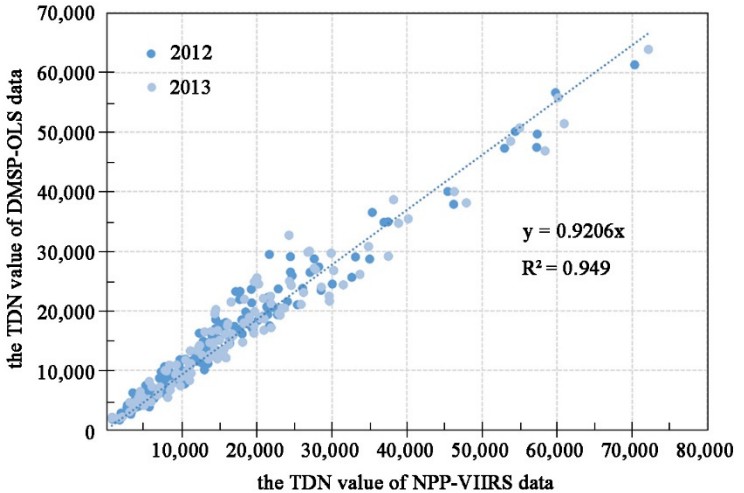

**Figure 3.** Relationship between the DMSP-OLS and NPP-VIIRS data for 2012 and 2013.

DMSP-OLS data (2005–2013) and NPP-VIIRS data (2012–2020) were obtained from the National Oceanic and Atmospheric Administration's National Geophysical Data Center (NOAA/NGDC) website (https://www.ngdc.noaa.gov/eog/download.html, accessed on 1 April 2021). The energy consumption data and coefficients for all types of fossil energy were derived from the China Energy Statistical Yearbook (2005–2020).

*3.2. Quantifying Urban Form*

To quantify urban form, landscape metrics were selected to describe two aspects of urban form: compactness and complexity (Table 1). In science and policy-making, the compactness urban development strategy remains a controversial issue [65]. Although a reasonable level of urban compactness may reduce private car dependency, control urban sprawl, and protect green spaces, it may still cause various socio-environmental problems when it goes beyond a certain level [28,66], such as traffic congestion, urban heat islands, the degradation of air quality, and the increase of greenhouse gas emissions. At the same time, urban complexity has attracted considerable interest and has been a focus of recent research on sustainable urban forms. In addition, these four landscape metrics were chosen mainly because they have been widely used and have proven to be effective for describing urban compactness and complexity [67] and also because they do not correlate with each other.

**Table 1.** Description of landscape metrics.

| Indicators | Abbreviation | Equation | Description |
|---|---|---|---|
| Patch cohesion index (units: %) | COHESION | $\text{COHESION} = \left(1 - \frac{\sum_{i=1}^{m}\sum_{j=1}^{m} p_{ij}^*}{\sum_{i=-1}^{m}\sum_{j=1}^{n} p_{ij}^* \sqrt{a_{ij}^*}}\right)\left(1 - \frac{1}{\sqrt{z}}\right)^{-1}(100)$ | $p_{ij}^*$ = perimeter of patch *ij* in terms of number of cell surfaces. $a_{ij}^*$ = area of patch *ij* in terms of number of cells Z = total number of cells in the landscape. |
| Percentage of like adjacencies (units: %) | PLADJ | $\text{PLADJ} = \left(\frac{\sum_{i=1}^{m} g_{ii}}{\sum_{i=1}^{m}\sum_{k=1}^{n} g_{ik}}\right)(100)$ | $g_{ii}$ = number of like adjacencies between pixels of class *i* based on the double-count method. $g_{ik}$ = number of adjacencies between class *i* and *k* based on the double-count method. |
| Landscape shape index (units: none) | LSI | $\text{LSI} = \frac{0.25\sum_{k=1}^{m} e_{ik}^*}{\sqrt{TA}}$ | $e_{ik}^*$ = total length of edge in landscape between classes *i* and *k*. TA = total landscape area (m$^2$). |
| Mean patch shape index (units: none) | SHAPE_MN | $\text{SHAPE\_MN} = \frac{\sum_{i=1}^{n} \frac{p_{ij}}{min p_{ij}}}{n_i}$ | $p_{ij}$ = perimeter of patch *ij* in terms of number of cell surfaces. min$p_{ij}$ = minimum perimeter of patch *ij* in terms of number of cell surfaces. |

Urban compactness reflects the degree of dispersion or aggregation of an urban area. The landscape metrics COHESION and PLADJ are the major indicators for characterizing urban compactness. The patch cohesion index (COHESION) measures the physical connectedness of urban land patches [68]. The value of COHESION is bound between 0 and 100. COHESION increases as the patch distribution becomes more clumped or aggregated and thus more physically connected [69]. The percentage of like adjacencies (PLADJ) quantifies the contiguity of urban development by evaluating the number of like adjacencies (i.e., urban pixels and other adjacent urban pixels) [70]. The values range between 0 and 100. The higher the PLADJ value, the more concentrated the urban land patches are.

Urban complexity reflects the extent of irregularity of the surface geometry of a specific patch type (e.g., the shape of an urban build-up patch). The landscape metrics LSI and SHAPE_MN are the major indicators for quantifying urban complexity. The landscape shape index (LSI) is a ratio of the perimeter to the area of the urban land patch [71]. The higher the LSI value, the more complex the shape. Similarly, the mean patch shape index (SHAPE_MN) is another metric for quantifying urban form complexity [72]. If an urban area has a high SHAPE_MN value, its shape is more irregular and complex.

Urban land-use data are essential for calculating landscape metrics to represent urban forms. In this study, considering the development speed of small and medium-sized cities, land-use data from four years (2005, 2010, 2015, and 2020) were used to estimate the urban form from 2005 to 2020. The land-use data were obtained from the Institute of Geographical Sciences and Natural Resources Research of the Chinese Academy of Sciences (CAS). Land-use data were interpreted from Landsat TM/ETM with a spatial resolution of 30 × 30 m. The boundary shapefiles of cities were derived from the National Geomatics Center of China. The above calculations were performed by using Fragstats4.2 software.

*3.3. Measuring Socioeconomic Factors*

This study used the industrial structure, population density, and economic development level to characterize socioeconomic factors, which are widely thought to significantly influence urban $CO_2$ emissions [73–75]. To analyze whether and how the UF–$CO_2$ relationship varies across cities with different socioeconomic characteristics, the 132 cities were categorized based on the above three aspects (Table 2). Socioeconomic data were collected from the City Statistical Yearbook (2005–2020) of 132 cities. The Jenks optimal natural fracture method in ArcGIS10.2 was adopted to divide the population density and economic development level [76], which minimized the sum of internal variances at all levels and excluded the influence of human factors as much as possible.

**Table 2.** The categories of cities based on three socioeconomic characteristics.

| Socioeconomic Characteristics | Types | Description | Amount |
|---|---|---|---|
| Industrial structure (IP) (units: %) | Industrial cities | GDP (secondary industry) $\geq$ 50% | 45 |
| | Service cities | GDP (tertiary industry) $\geq$ 50% | 36 |
| | Industrially balanced cities | Relatively balanced proportions of secondary and tertiary industries | 51 |
| Population density (POD) (units: person/KM2) | Low population density cities | <500 | 30 |
| | Medium population density cities | 500–1500 | 66 |
| | High population density cities | >1500 | 36 |
| Economic development level (PGDP)*Per capita GDP (units: 10,000 yuan) | Low-development-level cities | <5 | 47 |
| | Medium-development-level cities | 5–9 | 49 |
| | High-development-level cities | >9 | 36 |

*3.4. Panel Data Modeling*

To quantify the differential impacts of the socioeconomic factors (three socioeconomic factors) on the UF–$CO_2$ relationship, panel data models were used in this study. This method was chosen mainly because panel data models usually involve more observations than time-series and cross-sectional data sets, which can help reduce the effects of multicollinearity. This characteristic generally gives them a higher degree of freedom than cross-sectional models and reduces the collinearity of explanatory variables [77,78], which can decrease data bias to a greater extent. Furthermore, panel data models can describe the dynamic adjustment process of research objects and explain the relationships between variables more accurately [12]. Compared to conventional cross-sectional and time-series models, panel data models have obvious advantages.

To gain an overall understanding of the impact of urban form on $CO_2$ emissions under different socioeconomic characteristics, two types of regression models were established. First, interaction effect assumptions were made that the UF–$CO_2$ relationship depends on industrial structure, population density, and economic development level. Thus, regression models with interaction terms were established (Equation (1)) to explore whether socioeconomic factors have an impact on the UF–$CO_2$ relationship. The interaction terms were expressed as a landscape metric $\times$ a socioeconomic factor. When the correlation coefficient of the interaction term was positive, it means that the socioeconomic factor increased the impact of urban form on $CO_2$ emissions. When the correlation coefficient of the interaction term was negative, it means that the socioeconomic factor decreased the impact of urban form on $CO_2$ emissions. Second, to further evaluate the impact of urban form on $CO_2$ emissions, sub-sample regression models were established (Equation (2)).

$$y_{it} = \sum_{i=1}^{n} \beta_{it} x_{it} + \sum_{i=1}^{n} (\beta_{it} x_{it} \times x_{it\_m}) + u_{it} \tag{1}$$

$$y_{it} = \sum_{i=1}^{n} \beta_{it} x_{it} + u_{it} \tag{2}$$

where $y_{it}$ represents the per capita $CO_2$ emissions of city $i$ in year $t$; $x$ represents the landscape matrices (COHESION, PLADJ, SHAPE_MN, and LSI); $x_{it\_m}$ is a potential factor with interactive effects on urban form, in this study, including industrial structure (IP), population density (POD), and economic development level (PGDP); and $u_{it}$ is the random error. Both the dependent and independent variables underwent natural logarithmic transformations to reduce the heteroscedasticity [79].

The panel data analysis in this study can be summarized in three steps. First, the panel unit root tests and panel co-integration tests were performed to examine the stationary nature of time-series variables and whether a long-run relationship existed between the variables. If the model was established for nonstationary data despite having a high

correlation coefficient value, spurious regression can easily appear, and the results could be meaningless. Since the sampled period in this study is short, the unit root test proposed by Harris and Tzavalis (*HT*-test) is more appropriate for assessing the stationarity of the variables [80]. If all variables reject the null hypothesis, the panel data are considered stationary, and regressions can be performed. If the null hypothesis is accepted, the Pedroni panel co-integration tests are employed [81]. Then, the *F*-test and Hausman test are used to determine which specific regression form and effect form should be chosen. The *F*-test is used to identify whether the pooled regression model, constant coefficients and variable intercepts model or variable coefficients and variable intercepts model should be selected. The Hausman test was applied to identify whether the fixed or random-effects model should be selected. Finally, according to the *F*-test and Hausman test results, all models were estimated to establish models to evaluate the effects of urban form on $CO_2$ emissions. The above analysis was performed using Stata/SE software.

## 4. Results

### 4.1. Variations in Urban Form and $CO_2$ Emissions

From 2005 to 2020, small and medium-sized cities in the YRD experienced a series of significant changes in urban form. Figure 4 shows the statistical characteristics of the landscape metrics for 132 cities for 2005 and 2020. In terms of the urban compactness, the mean patch cohesion index (COHESION) and the mean percentage of like adjacencies (PLADJ) showed an increasing trend in 2020 compared with 2005, indicating a more aggregated urban form and the tendency for urban patches to gradually become spatially connected. Furthermore, the interval between the maximum and minimum COHESION was significantly smaller in 2020, indicating that the difference in compactness among small and medium-sized cities was reduced. Regarding the complexity of the urban form, the mean patch shape index (SHAPE_MN) increased by 12.17%, from 1.430 in 2005 to 1.604 in 2020, and the mean landscape shape index (LSI) also increased slowly. The trend of the two indexes suggests that the urban form of small and medium-sized cities became fragmented and irregular and that spatial heterogeneity gradually increased with urban development.

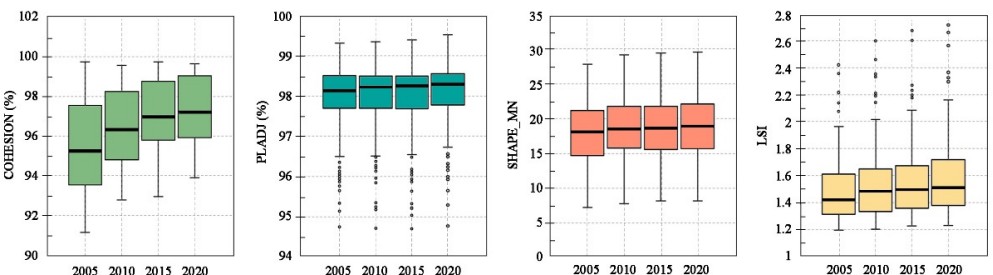

**Figure 4.** Changes in the landscape metrics of 132 cities in the YRD, 2005 to 2020.

This study used per capita $CO_2$ emissions (PCO$_2$) as an indicator to measure the level of $CO_2$ emissions instead of the total amount of $CO_2$ emissions. This is mainly because PCO$_2$ can reflect the level of regional aggregate $CO_2$ emissions and the level of regional socio-civic development while avoiding the impact of urban area size on the $CO_2$ emissions [82]. The PCO$_2$ of 132 small and medium-sized cities in the YRD from 2005 to 2020 was calculated based on the $CO_2$ emissions estimation data and statistical population data (Table 3). During the research period, the PCO$_2$ of small and medium-sized cities generally showed a continuous increase. The average PCO$_2$ increased from 3.714 t in 2005 to 4.855 t in 2020 (an increase of 30.72%), peaked in 2015 at 4.951 t, and decreased slightly from 2015 to 2020. At the same time, the standard deviation of PCO$_2$ increased from 2.946 in 2005 to 3.769 in 2020, indicating that the difference in energy consumption and development levels among cities rose.

**Table 3.** Statistics of PCO$_2$ for 132 cities in the YRD, 2005 to 2020.

| Variable | Mean | Std.Dev. | Min | Max | Median |
|---|---|---|---|---|---|
| 2005 | 3.714 | 2.946 | 0.198 | 16.412 | 3.09 |
| 2010 | 4.840 | 3.784 | 0.383 | 20.672 | 4.10 |
| 2015 | 4.951 | 3.846 | 0.422 | 20.731 | 4.19 |
| 2020 | 4.855 | 3.769 | 0.376 | 20.265 | 4.17 |

Min represents minimum; Max represents maximum; Std.Dev. represents standard deviation.

### 4.2. Interaction Effects between Socioeconomic Factors and Urban Form

Model 1 revealed the correlation between urban form and PCO$_2$ for 132 small and medium-sized cities in the YRD to determine whether urban form has an impact on PCO$_2$ in small and medium-sized cities. Models 2–4 are regression models with interaction terms that were used to estimate the relationship between urban form and PCO$_2$ under three socioeconomic factors. Before conducting panel data models, a series of model-related tests were performed. First, we performed the panel unit root tests (*HT*-test) (Table 4). The results suggested that the levels of the variables rejected the null hypothesis of nonstationarity at a significance level of 1%, indicating that all variables were stationary. Therefore, panel data modeling could be established directly to estimate the impacts of the dependent variable on the independent variable without performing the panel co-integration test further.

**Table 4.** Results of panel unit root tests.

| Variable | Model 1 | Model 2 | Model 3 | Model 4 |
|---|---|---|---|---|
| per-capita CO$_2$ emissions (PCO$_2$) | −8.180 *** | −8.180 *** | −8.180 *** | −8.180 *** |
| COHESION | −3.480 *** | −3.480 *** | −3.480 *** | −3.480 *** |
| PLADJ | −12.057 *** | −12.057 *** | −12.057 *** | −12.057 *** |
| SHAPE_MN | −10.661 *** | −10.661 *** | −10.661 *** | −10.661 *** |
| LSI | −8.399 *** | −8.399 *** | −8.399 *** | −8.399 *** |
| IP | | −11.636 *** | | |
| POD | | | −3.846 *** | |
| PGDP | | | | −0.667 * |
| COHESION × IP | | −11.639 *** | | |
| PLADJ × IP | | −11.642 *** | | |
| SHAPE_MN × IP | | −10.979 *** | | |
| LSI × IP | | −12.001 *** | | |
| COHESION × POD | | | −3.778 *** | |
| PLADJ × POD | | | −3.852 *** | |
| SHAPE_MN × POD | | | −9.539 *** | |
| LSI × POD | | | −3.924 *** | |
| COHESION × PGDP | | | | −0.750 * |
| PLADJ × PGDP | | | | −0.654 * |
| SHAPE_MN × PGDP | | | | −8.331 *** |
| LSI × PGDP | | | | −3.370 *** |

*, and *** indicate significance at 10%, and 1%, respectively.

To determine the specific regression and effect forms, the *F*-test and Hausman test were performed successively. The *F*-test results for all the models are presented in Table 5. The test results of the four models were consistent, and the F-statistic was greater than the critical value at the 5% significance level. This means that variable intercepts were adopted for these models. Next, the Hausman test was performed on the four models to compare the reliability between the fixed or random-effects models. The null hypothesis of the Hausman test is that the random-effects model is more suitable. Table 6 presents the results of the Hausman test. The *p*-values of the four models were greater than the critical value at the 5% significance level. That is, the null hypothesis of random-effects was rejected. Thus, four models were established based on the fixed-effects models.

**Table 5.** Results of *F*-test.

| Model | *F*-Test |
|-------|----------|
| Model 1 | F (38.955) > $F_{0.05}$ (1.253) |
| Model 2 | F (34.806) > $F_{0.05}$ (1.254) |
| Model 3 | F (32.883) > $F_{0.05}$ (1.254) |
| Model 4 | F (33.467) > $F_{0.05}$ (1.254) |

**Table 6.** Results of Hausman test.

| Model | Chi-Sq Statistic | Prob. |
|-------|------------------|-------|
| Model 1 | 29.86 | 0.000 |
| Model 2 | 44.00 | 0.000 |
| Model 3 | 11.99 | 0.062 |
| Model 4 | 30.83 | 0.000 |

Table 7 displays all the estimation results between the landscape metrics and $PCO_2$ of 132 small and medium-sized cities in the YRD. It can be found that the correlation coefficients ($R^2$) of Models 2, 3, and 4 are 0.593, 0.615, and 0.707, respectively, which were significantly greater than that of Model 1 (0.554). All the models for Model 2 to Model 4 have better performance in estimation, indicating that the three socioeconomic factors chosen in this study will all have an impact on the relationship between urban form and $PCO_2$. In other words, considering socioeconomic characteristics can improve the accuracy of estimating the relationship between urban form and $PCO_2$.

**Table 7.** Results of the panel data model estimation.

| Independent Variable | Dependent Variable: Per-Capita CO$_2$ Emissions (PCO$_2$) | | | |
|---|---|---|---|---|
| | **Model 1** | **Model 2** | **Model 3** | **Model 4** |
| COHESION | −7.041 ** | −7.056 *** | −17.068 *** | −12.270 *** |
| PLADJ | −9.365 * | −11.623 * | −26.261 *** | −23.428 *** |
| SHAPE_MN | 1.253 *** | −0.254 | 1.041 *** | −0.517 |
| LSI | 1.628 *** | 0.615 | 0.776 * | 0.108 |
| IP | - | 32.019 ** | - | - |
| POD | - | - | 15.778 * | - |
| PGDP | - | - | - | 21.040 *** |
| COHESION × IP | - | −3.091 *** | - | - |
| PLADJ × IP | - | −3.081 *** | - | - |
| SHAPE_MN × IP | - | 0.023 | - | - |
| LSI × IP | - | 0.013 | - | - |
| COHESION × POD | - | - | 2.533 *** | - |
| PLADJ × POD | - | - | 2.364 *** | - |
| SHAPE_MN × POD | - | - | 0.201 ** | - |
| LSI × POD | - | - | 0.175 *** | - |
| COHESION × PGDP | - | - | - | 1.050 ** |
| PLADJ × PGDP | - | - | - | 1.061 *** |
| SHAPE_MN × PGDP | - | - | - | 0.045 |
| LSI × PGDP | - | - | - | 0.007 |
| Constant | 10.739 | 21.054 | −20.172 | −177.913 ** |
| N | 528 | 528 | 528 | 528 |
| $R^2$ | 0.554 | 0.593 | 0.615 | 0.707 |

*, **, and *** indicate significance at 10%, 5%, and 1%, respectively.

In Model 1, all the landscape metrics were significantly correlated with $PCO_2$. In terms of compactness, COHESION and PLADJ were negatively correlated with $PCO_2$ (−7.041, −9.365), indicating that increasing the compactness of small and medium-sized cities has a positive impact on reducing $PCO_2$. When controlling for other factors, $PCO_2$ was reduced

by 7.041% and 9.365% for each 1% increase in COHESION and PLADJ, respectively. In terms of complexity, SHAPE_MN and LSI were significantly positively correlated with $PCO_2$ (1.253, 1.628), indicating that $PCO_2$ showed a declining trend as the complexity of the patch shapes decreased. $PCO_2$ was reduced by 1.253% and 1.628% for each 1% decrease in SHAPE_MN and LSI, respectively. However, compactness has a stronger ability to regulate $PCO_2$. Overall, compactness and low complexity urban development patterns are more conducive to reducing the $PCO_2$ of small and medium-sized cities in the YRD.

In Model 2, when considering the potential effect of industrial structure, the variable $PCO_2$ was negatively correlated with compactness. Under the control of other unchanged conditions, for each 1% increase in COHESION and PLADJ, $PCO_2$ decreased by 7.056% and 11.623%, respectively. Meanwhile, the interaction term (COHESION × IP, PLADJ × IP) results suggested a significant negative interactive effect ($-3.091$, $-3.081$) between industrial structure and compactness. Increasing the IP value reduced the negative impact of compactness on $PCO_2$. That is, when the proportion of secondary industry in the city is higher, the inhibition effect of compactness on $PCO_2$ is less significant.

Model 3 analyzed the impact of population density on the relationship between urban form and $PCO_2$. The results showed that urban compactness and complexity have opposing effects on the variable $PCO_2$. The COHESION ($-17.068$) and PLADJ ($-26.261$) both had a significant negative impact on $PCO_2$, while SHAPE_MN (1.041) and LSI (0.776) were positively correlated with $PCO_2$. The increase in compactness and reduction in complexity were more conducive to a lower $PCO_2$. At the same time, population density shows positive interactive effects (2.533 and 2.364, 0.201 and 0.175) with urban compactness (COHESION × POD, PLADJ × POD) and complexity (SHAPE_MN × POD, LSI × POD), respectively, meaning that the higher the population density of cities, the more significant the impact of urban compactness and complexity on $PCO_2$.

Model 4 analyzed the impact of urban form on $PCO_2$ when considering the role of economic development level. The results showed that urban compactness was negatively correlated with $PCO_2$. For each 1% increase in COHESION and PLADJ, the $PCO_2$ was reduced by 12.270% and 23.428%, respectively. Increasing urban compactness significantly curbed $PCO_2$. Meanwhile, a significant positive interactive effect (1.050, 1.061) between per capita GDP and compactness (COHESION × PGDP, PLADJ × PGDP) suggests that the higher the level of economic development of a city, the more pronounced the suppressive effect of urban compactness on $PCO_2$.

### 4.3. Impact of Socioeconomic Factors on the Relationship between Urban Forms and $CO_2$ Emissions

As presented in Table 7, the results explain two key issues: whether there are interaction effects between socioeconomic factors and urban form for small and medium-sized cities in the YRD and whether socioeconomic characteristics weaken or enhance the impact of urban form on $PCO_2$. However, for small and medium-sized cities with different industrial structures, population densities, and economic development levels, the effects of urban form on $PCO_2$ have not been clarified. Therefore, a sub-sample regression was used for further analysis based on the results of the significant correlations in Table 6 (Models 2 to 4). The analysis results are shown in Tables 7–9.

Table 8 shows the relationship between urban form and $PCO_2$ in small and medium-sized cities with different industrial structures. Specifically, neither compactness metric for industrial cities passed the significance test, which suggests that changes in the urban form do not significantly impact $PCO_2$ in industrial cities. For service cities and industrially balanced cities, an increase in compactness positively impacted the reduction in $PCO_2$; however, compactness had almost the same impact on $PCO_2$ in the two types of cities, and there was no significant difference between them.

As shown in Table 9, although the impact of urban compactness and complexity on $PCO_2$ gradually increased with population density, there were obvious differences in the impact of urban form on $PCO_2$ for cities with different population densities. Specifically,

for cities with high population density, $PCO_2$ could be reduced by 12.367% for each 1% increase in compactness (PLADJ). By contrast, in cities with low population density, the data would be reduced to 3.999%, and the adjustment ability of urban form to $PCO_2$ would be significantly reduced. Similarly, the landscape metric SHAPE_MN for characterizing complexity also showed a similar pattern. In cities with low population density, for each 1% increase in complexity, $PCO_2$ only increases by 0.040%, which shows little adjustment ability regarding $PCO_2$. In addition, compared with the complexity of the urban form, compactness has a more significant ability to regulate $PCO_2$.

**Table 8.** The relationship between urban form and $PCO_2$ under different industrial structures.

| Independent Variable | Dependent Variable: Per-Capita $CO_2$ Emissions ($PCO_2$) | | |
| --- | --- | --- | --- |
| | **Industrial Cities** | **Industrially Balanced Cities** | **Service Cities** |
| lnCOHESION | 4.799 | −5.974 *** | −5.981 *** |
| lnPLADJ | −23.371 | −45.897 * | −47.314 *** |
| Constant | 86.748 | 194.024 | 90.501 ** |
| N | 176 | 175 | 177 |
| $R^2$ | 0.303 | 0.513 | 0.329 |

*, **, and *** indicate significance at 10%, 5%, and 1%, respectively.

**Table 9.** The relationship between urban form and $PCO_2$ under different population densities.

| Independent Variable | Dependent Variable: Per-Capita $CO_2$ Emissions ($PCO_2$) | | |
| --- | --- | --- | --- |
| | **Low Population Density** | **Medium Population Density** | **High Population Density** |
| lnCOHESION | −32.587 | −3.693 | −6.512 |
| lnPLADJ | −3.999 *** | −6.403 *** | −12.367 *** |
| lnSHAPE_MN | 0.040 *** | 0.352 *** | 0.596 *** |
| lnLSI | 0.732 | −0.726 | 0.235 * |
| Constant | 130.524 | −12.587 | −23.853 |
| N | 176 | 176 | 176 |
| $R^2$ | 0.610 | 0.670 | 0.680 |

*, and *** indicate significance at 10%, and 1%, respectively.

As shown in Table 10, the relationship between urban form and $PCO_2$ changed gradually with various economic development levels in small and medium-sized cities. For cities with a medium economic development level and high economic development level, increasing the compactness had a positive impact on reducing $PCO_2$, especially for cities with high economic development levels. When controlling for other conditions, the $PCO_2$ of cities with high economic development levels was reduced by 65.032% for each 1% increase in compactness (PLADJ). However, for cities with low economic development levels, neither indicator passed the significance test. This indicates that the urban form has no impact on $PCO_2$ in cities with low economic development levels.

**Table 10.** The relationship between urban form and $PCO_2$ under different economic development levels.

| Independent Variable | Dependent Variable: Per-Capita $CO_2$ Emissions ($PCO_2$) | | |
| --- | --- | --- | --- |
| | **Low Development Level** | **Medium Development Level** | **High Development Level** |
| lnCOHESION | −1.615 | −0.036 | −7.209 ** |
| lnPLADJ | 19.947 | −17.205 ** | −65.032 ** |
| Constant | −81.948 | 80.418 ** | 266.783 |
| N | 179 | 173 | 176 |
| $R^2$ | 0.617 | 0.691 | 0.649 |

** indicate significance at 5%.

## 5. Discussion

From the results of the interaction effects analysis and sub-sample regression analysis, several meaningful findings on the relationships between the urban spatial form and $PCO_2$ of small and medium-sized cities can be derived.

Comprehensive research results showed that a compact urban form in small and medium-sized cities was more favorable for the realization of $CO_2$ emission reductions. The findings of this study support the theory that having a more compact city may affect transportation $CO_2$ emissions by influencing travel frequency and mode choice while reducing heating-associated energy consumption in buildings by affecting the heat island effect [83,84]. However, in contrast to the high-density and intensive urban land development strategy of megacities and big cities, most of China's small and medium-sized cities are expanding at a low density [85–87]. The natural space (forest, grassland, and agricultural land) around small and medium-sized cities is more abundant, resulting in relatively low urban heat island effects [88,89]. The research on the relationship between the heat island effect and building energy consumption in small and medium-sized cities in China has not attracted much attention. At the same time, it has been proven that the low-density expansion of small and medium-sized cities can lead to higher $PCO_2$ emissions from transportation [90]. Therefore, we speculate that the compactness of the spatial form of small and medium-sized cities mainly affects the transportation sector's energy consumption and $CO_2$ emissions [91]. The main reason for the negative correlation between $PCO_2$ and complexity is that the irregular and complex urban build-up patch significantly increased the distance and duration of automobile trips in daily life when living and working areas are distributed in different urban patches [30,92]. Meanwhile, a more complex and fragmented urban form will intensify investment in infrastructure construction, such as road systems, water supplies, and drainage pipe networks, thereby increasing $CO_2$ emissions [93].

By comparing the results of small and medium-sized cities with different industrial structures, we found that the $PCO_2$ of service cities and industrially balanced cities are affected by compactness. Several aspects may explain this. For small and medium-sized cities, when the proportion of tertiary industry reaches a certain level, the scale and intensity of urban expansion are higher. Additionally, along with a large number of vehicles and population, the phenomenon of energy consumption in the transportation sector is evident, with high connectivity and an agglomerated urban form being more favorable for reducing residents' $CO_2$ emissions in their daily commute. Second, compared to the low-development-level cities, in relatively developed small and medium-sized cities, the phenomenon of the unequal allocation of resources between the main urban areas and the suburban areas is usually more prominent. More high-quality medical and educational resources gather in the main urban areas, leading to residents in suburban areas and new districts needing to travel longer distances to access them. Compact development may result in lower interzonal interactions [80]. Therefore, a more compact urban form has a more significant intervention effect on the $PCO_2$ in service and industrially balanced cities. The empirical results for industrial cities are consistent with the previous literature [15]. Industrial activities are usually a major source of $CO_2$ emissions in Chinese cities. However, they are mainly closely related to technical conditions, light and heavy industrial structures, and energy structures while not being significantly correlated with urban compactness.

By comparing cities with different population densities, we found that compactness and complexity affect the $PCO_2$ of all types of small and medium-sized cities, and the impacts of both compactness and complexity on $PCO_2$ were more significant with an increase in population density. The results of urban compactness and $PCO_2$ can be explained by the fact that most of the small and medium-sized cities in the YRD are mononuclear cities, which generally have function-concentrated characteristics with education, medical care, commerce, leisure, culture, and financial support [94]. That is, the degree of land-use mixing in central urban areas is usually high. The high land use mixing here mainly refers to the combination and mix of various functions rather than high building density [95,96],

as urban economic growth strategies such as land finance have led to the low-density expansion of small and medium-sized cities in recent decades. The fuel consumption for transportation in the daily movement of individuals can be significantly reduced by providing opportunities for living, working, and entertainment activities in the same urban area and location (within an urban land patch or several adjacent urban land patches). Thus, the higher the population density, the easier it is to exert the beneficial agglomeration effect of compact cities, and the more obvious the positive effect of highly mixed land use on reducing $CO_2$ emissions [29]. Second, the positive impacts of urban form complexity on $PCO_2$ were more significant with an increase in population density, which is consistent with previous studies [28]. This may be because, on the one hand, the increase in population density inevitably results in increases in household and transportation energy consumption [28]. On the other hand, the increase in population density results in an increase in residential and infrastructure land, thereby leading to a reduction in the resources of vegetation coverage and carbon storage [7,67]. Overall, for cities with a high population density, $PCO_2$ was more sensitive to urban complexity.

Among cities with different levels of economic development, we found that the inhibition effect of compactness on $PCO_2$ was most significant in cities with high economic development levels, followed by cities with medium economic development levels. However, compactness had no significant impact on cities with a low level of economic development. The possible reasons for this are mainly the differences in residents' lifestyles and energy consumption patterns. In small and medium-sized cities with high economic development levels, the residents' income level is high, which contributes to an increase in vehicle ownership and the utilization rate of private cars [97,98]. In Chinese socioeconomic culture, private vehicles are regarded as a symbol of social status [83,99]. Meanwhile, the gathering of high-quality public resources is more obvious in cities with high economic development levels, and small and medium-sized cities generally suffer from poorly developed public transport systems, resulting in a high proportion of residents using private cars for long-distance travel [90]. Therefore, the $PCO_2$ in cities with high and medium economic development levels are more sensitive to urban compactness. The results are also mutually confirmed to a certain extent using the above analysis of the industrial structure. In addition, it is easy to speculate that the continued growth in GDP and further enhancement of residential income were the main driving factors for the growth of $PCO_2$ from transportation in small and medium-sized cities, which is consistent with previous studies [100].

This study has several limitations that need to be addressed in future research. First, it was limited by the available data from the Chinese statistical system, which explores only three socioeconomic factors: industrial structure, population density, and economic development level. More socioeconomic characteristics should be considered in follow-up research to provide a more comprehensive understanding of the UF–$CO_2$ relationship. Second, as the detailed composition (i.e., transportation, building, and industry) of the energy consumption is not fully available, future research could solve this problem through data mining of multisource data, energy consumption simulations, and predictions in order to achieve a more specific interpretation of the effect mechanism of urban form on $CO_2$ emissions. Third, as the disparities across small and medium-sized cities in different regions are significant, more cross-regional research and horizontal comparisons should be conducted in the future.

## 6. Conclusions

This study explores the impact of urban form on per capita $CO_2$ emissions under different socioeconomic factors (industrial structures, population densities, and economic development levels) in 132 small and medium-sized cities in the Yangtze River Delta. Five main conclusions can be drawn:

(1) Per-capita $CO_2$ emissions increased in small and medium-sized cities in the YRD between 2005 and 2020 but showed a slight declining trend after 2015.

(2)   The patch cohesion index, percentage of like adjacencies, landscape shape index, and mean patch shape index increased, demonstrating that the urban form of the small and medium-sized cities in this study became more compact and complex.

(3)   The comparison of the three socioeconomic factors showed that the relationship between urban form and per-capita $CO_2$ emissions was more significantly affected by the level of economic development, followed by population density and industrial structure.

(4)   From the analysis of interactive effects, we found that the negative impact of compactness on per-capita $CO_2$ emissions increased as the proportion of the tertiary industry, population density, and per-capita GDP rose. The positive impact of complexity on per-capita $CO_2$ emissions only increased with a rise in population density.

(5)   The sub-sample regression analysis demonstrates that the impact of compactness on per-capita $CO_2$ emissions was significant in service cities and industrially balanced cities, while there was no obvious correlation in industrial cities. Urban form compactness and complexity had a significant correlation with $PCO_2$ in all cities with different population densities, and the levels both rose with an increase in population density. The impact of the urban compactness on per-capita $CO_2$ emissions was most significant in cities with a high development level, followed by medium-development-level cities, while the impact was insignificant in low-development-level cities.

The findings in this study suggest that urban planning policies could be a key measure for alleviating the problem of increased $CO_2$ emissions in small and medium-sized cities. Meanwhile, these approaches should vary from region to region because of the differentiated effect levels on the relationship between urban form and $CO_2$ emissions under different socioeconomic factors.

(1)   Policies and planning strategies should aim to increase urban compactness, especially for small and medium-sized cities with a higher proportion of the tertiary industry, higher population density, and higher per capita GDP. Small and medium-sized cities should promote urban growth via infilling rather than outlying. This is also consistent with the land-use policies of cities in the YDR in recent years [101,102]. Decision-makers should seek to redevelop underutilized land and brownfields within cities to reduce mobility requirements and vehicle dependency by increasing efficient land use and combining the benefits of the centralization of various functions. In this process, the occupation of natural space should be avoided and interspersed with blue–green spaces as much as possible to counteract the negative effects of highly compact cities (i.e., urban island effects and lower quality of life) [103].

(2)   Consummating urban public transportation networks, especially between the main urban areas and suburban areas, is a method of compensating for the long-distance travel problem of private cars caused by the unbalanced allocation of public resources.

(3)   Minimizing complexity should be regarded as a supplementary method for optimizing urban form to mitigate $CO_2$ emissions. However, removing and modifying urban built-up patches is unrealistic and expensive. Thus, decreasing urban complexity by adjusting the road system (i.e., adding or removing urban branch roads and one-way streets) to avoid unnecessary vehicle detours whenever possible may be a more economically feasible way to reduce the negative effects on transport $CO_2$ emissions caused by urban complexity.

(4)   For industrial cities, although the role of urban form in reducing $CO_2$ emissions is not as significant as it is in other cities, the overall $CO_2$ emissions of industrial cities are usually high. Therefore, $CO_2$ emission reduction in industrial cities can be implemented not only by improving energy efficiency and adjusting the industrial structure but also by long-term spatial optimization measures, such as improving compactness and reducing complexity.

**Author Contributions:** Conceptualization, resource preparation, data analysis, and writing—original draft, R.G.; methodology, software, validation, and visualization, R.G. and S.S.; supervision, project

administration, and funding acquisition, Q.Y. and H.L. All authors have read and agreed to the published version of the manuscript.

**Funding:** This work was funded by the National Key Research and Development Project, grant number 2018YFC0704705.

**Institutional Review Board Statement:** Not applicable.

**Informed Consent Statement:** Not applicable.

**Data Availability Statement:** Not applicable.

**Acknowledgments:** We greatly appreciate Vahid M. Nik of Lund University for his constructive comments, which improved the quality of the paper. We would also like to thank the Division of Building Physics, Department of Building and Environment Technology, Lund University for their support and for providing us with the necessary facilities.

**Conflicts of Interest:** The authors declare no conflict of interest.

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
