# Peer review of "Impact of Urban Form on CO2 Emissions under Different Socioeconomic Factors: Evidence from 132 Small and Medium-Sized Cities in China"

_land, doi:10.3390/land11050713_

Round 1
Reviewer 1 Report
The study aims to quantify the impact of urban compactness and urban complexity on CO2 emissions for a large number of medium and small cities in China while taking into account socio-economic factors. The topic is highly relevant and the authors are presenting a detailed statistical methodology to achieve a rather complex quantitative assessment. While some interesting findings are listed, the manuscript requires major revision to make it more accessible to the reader. Specifically,
- it should be made more clear in the introduction which open questions in the current literature are being addressed by the work presented. Then in the conclusions, the authors should refer to these clear points to highlight where actual novel insights are obtained. Specifically it should be highlighted where findings are not only confirming the obvious (increased urban density reduces emissions due to reduced heating demands and shorter transit). Strengthen the NOVEL findings or explain where your findings may contradict previous studies.
- The authors are not using an evaluated emissions inventory but rather obtain CO2 emission estimates from a combination of night-lights data. It needs to be clearly identified how potential uncertainties in the obtained CO2 emissions assessment may be linked to the processes studied here. Which emission sources are being accounted for? What is their relative uncertainty? E.g. are transport emissions captured with a similar accuracy as emissions from heavy industry? And how do the studied factors of e.g. urban density or complexity affect the CO2 emission estimate? It should be clearly outlined that the utilised data are suitable for exploitation in the current study.
- The methods and results are presented in a rather technical manner. Try to reduce the use of symbols and acronyms in the text to make it more easily accessible to the reader.
Minor comments:
Page2, line 51: What do you consider “conventional methods of reducing CO2”?
Page 2, line 52: Not clear what is the “improper impact of urban form on CO2 emissions”. It is important for the context of the manuscript that you start by explaining why you expect aspects of urban form (in addition to obvious urban characteristics such as population density, GDP, industry, etc) should impact CO2 emissions.
Page 2, line 53: What do you mean by “influence is usually rather slow”? not clear.
Page 2, line 54: I think you do not explain sufficiently why “urban form and CO2 emissions is critically important”.
Page 2, line 72: Please explain what are “mononuclear patterns, and multiple-nuclei patterns”.
Page 2, line 93: why is the household energy consumption increased if green spaces are further away? Is this related to cooling demand? This would suggest to a conclusion that can be highly dependant on latitude and or local climates. How are you taking these aspects into account in the design of your study?
Page 2, line 95: Looks like maybe the findings could be consistent but not fully conclusive, i.e. the UF-CO2 relation could be impacted by other factors such as availability of public transport, congestion and road traffic policies, geographic setting and background climate conditions, … Please demonstrate here in the introduction how your research approach will be able to advance the scientific conclusions in this field.
Page 3, line 112: State explicitly how you define small and medium sized cities, respectively.
Page 3, line 118: add climate conditions. Also, what about diversity in urban planning (public transport), industrialisation, and carbon emission reduction policies. All of which you mentioned are important for the UF-CO2 relation.
Page 3, line 151: growth rate in terms of what? Impervious surface?
Page 4, line 156: so from the figure it looks as if the entire YRD is either urban or water surface. What about the “do data” areas? Are there no natural spaces or agriculture?
Page 4, line 169: Please clarify what is the “IPCC method“?
Page 5, line 197: Please comment: how is the method (including both datasets and the interpolation) uncertainty affected by urban form? How are uncertainties in night-time light quantification affected by urban compactness and complexity (e.g. building height variability)? Do you expect and systematic effect on the uncertainties of derived CO2 emissions that could affect the subsequent analysis?
Page 5, line 208: Provide details – why is compactness considered a controversial issue? In what sense is this relevant here?
Page 6, line 210: You mentioned compactness and complexity but now you are talking about “four landscape metrics” – where are these defined?
Page 6, line 214-220: please rephrase. The parameters are not explained very well.
Page 7, line 239-250: rephrase into a text or change to bullet points or table
Page 7, line 254: more data than what alternative?
Page 7, line 260: Add sentence to introduce the equations.
Page 9, line 315-321: Provide more details. What is being assessed and why? How do the conclusions fit into the overall objectives?
Page 9, line 324: remind the reader about these models. What do they do? How are they set up? What are the objectives of the analysis presented in this section?
Page 11, line 367: Why do regular urban patterns reduce CO2 emissions?
page 11, line 384: is it really possible to differentiate the effects of “compactness” and “population density”?
Page 13, line 429: please rephrase in common words. What has to be done to effectively reduce emissions in cities with high economic development level?
Page 13, line 446: please check literature. Of course there is evidence of heat island effects also for smaller urban agglomerations.
Page 13, line 449: What about the cooling demand in summer?
Page 13, line 455: But are those emissions from infrastructure investments accounted for in your CO2 emissions inventory?
Page 14, line 484: Above you state that complexity increases emissions because travel routes are more complicated and hence reduce efficient traffic flow. Now you state integration of land uses is efficient. Is this not a contradiction?
Page 15, line 532: Avoid using such symbols in the conclusions. The conclusions should be mostly stand-alone and should not require additional details.
Page 16, line 583: Be careful: does this conclusion on industrial cities also hold when you look at the absolute reduction in carbon emissions? Maybe relative effects are small because the total emissions of industrial cities are generally high, but if the goal is to reduce carbon emission sin total for the entire YRD region, the absolute emission reduction should be the focus.
Reviewer 2 Report
Dear Authors,
Please address the following issues:
The manuscript is not well written and needs a significant revision to make it easier to read and follow.
The introduction doesn't provide the required background on the topic and the reason for the need for doing this study.
The methodology is not well described and the figures are not clear.
Some figures miss the axis title.
Different variables have been defined/used in the manuscript which makes it hard to follow the research findings (for example, table 3).
Good luck
Reviewer 3 Report
I suggest supporting the methodology and the results section with some diagrams able to show how the chosen indicators affect the CO2. Moreover, I suggest introducing into the discussion some alternatives to the classic Chinese city development, also studying not Chinese UF.
